# GRAPHFM: A GENERALIST GRAPH TRANSFORMER THAT LEARNS TRANSFERABLE REPRESENTATIONS ACROSS DIVERSE DOMAINS

## ABSTRACT

Graph neural networks (GNNs) are often trained on individual datasets, requiring specialized models and significant hyperparameter tuning due to the unique structures and features of each dataset. This approach limits the scalability and generalizability of GNNs, as models must be tailored for each specific graph type. To address these challenges, we introduce GRAPHFM, a scalable multi-graph pretraining approach designed for learning across diverse graph datasets. GRAPHFM uses a Perceiver-based encoder with learned latent tokens to compress domain-specific features into a shared latent space, enabling generalization across graph domains. We propose new techniques for scaling up graph training on datasets of different sizes, allowing us to train GRAPHFM on 152 distinct graph datasets, spanning 7.4 million nodes and 189 million edges. This allows us to study the effect of scale on pretraining across domains such as molecules, citation networks, and product graphs, and show that training on diverse datasets improves performance over single-source pretraining. Our results demonstrate that pretraining on diverse real and synthetic graphs enhances adaptability and stability, leading to competitive performance with state-of-the-art models across various node classification tasks. This approach reduces the burden of dataset-specific training and provides a single generalist model capable of performing across multiple diverse graph structures and tasks.

## 1 INTRODUCTION

Graphs are a fundamental data structure used across diverse fields such as biology, social networks, and recommendation systems (Hamilton et al., 2017). However, most graph neural network (GNN) architectures are designed in a highly specialized way, optimized for specific types of graphs. For example, architectures that work well on homophilic graphs, such as citation networks, often fail to generalize to heterophilic graphs, like certain social or biological networks, due to the differences in their topologies (Abu-El-Haija et al., 2019; Yan et al., 2022). This specialization leads to a fragmentation in model development, where the optimal architecture for one type of graph must be significantly altered or redesigned for another. As the use of GNNs grows across diverse applications, this piecemeal approach limits scalability and generalization, highlighting the need for a generalist model that can handle a wide variety of graph structures without manual tuning.

A core challenge in building a generalist graph model lies in integrating diverse graphs, each with unique topologies, node features, and sizes, while enabling knowledge transfer across them. Without a shared "vocabulary" for graph structures, models struggle to generalize effectively, as the differences between graph types hinder the transfer of learned patterns (Galkin et al., 2023). At the same time, recent advances in large-scale language models have shown that scaling up both model size and data diversity is essential for unlocking emergent capabilities and improving generalization across tasks (Wei et al., 2022). This makes scaling an equally critical factor in graph models. Pretraining on diverse graphs requires algorithms that can efficiently handle large, heterogeneous inputs, while ensuring the model can still capture robust, transferable patterns. Therefore, building a generalist graph model necessitates solutions that not only integrate diverse graph structures but also scale effectively, allowing the model to learn from vast, varied datasets without sacrificing performance.

In this work, we introduce GRAPHFM , a multi-graph pretraining framework aimed at addressing this gap. Instead of building specialized models for each graph type, GRAPHFM uses a Perceiver-based transformer encoder (Jaegle et al., 2021b) to create a shared latent space that abstracts away graph-specific details while preserving core structural properties. This enables the model to process a wide range of graph types within a unified framework, moving beyond the specialist architectures that dominate current GNN design. Our approach seeks to answer a key question: can pretraining on diverse, multi-graph datasets lead to effective generalization and transfer across unseen graphs?

When tested on a variety of homophilic and heterophilic datasets, we demonstrate that our model achieves performance comparable to all of the best baseline models, each of which is individually tuned for its respective dataset. Overall, we achieve the best rank when compared with these models, demonstrating that our approach has strong generalist performance. By combining datasets from biology, social networks, and recommendation systems, we show that our model can generalize across graphs with varying topologies and features, providing the flexibility that specialized models often lack. Moreover, our framework efficiently handles large mixtures of diverse graph datasets, leveraging distributed training techniques to manage graphs of different sizes and complexities.

Our results show that increasing both the scale of the model and the diversity of the training data leads to significant improvements in downstream performance on new, unseen graphs and node-level tasks. This demonstrates that it is indeed possible to train a generalist model on diverse graphs, which can effectively learn from and adapt to a wide range of graph types. In total, we pretrain on 152 distinct graph datasets, comprising over 7.4 million nodes and 189 million edges across a wide variety of graph types—an unprecedented number of different graph datasets in the literature. This extensive pretraining allows our model to capture and transfer knowledge across a broad spectrum of graph structures, showcasing the feasibility and advantages of building a unified model that generalizes well to unseen tasks.

The main contributions of this work are as follows:

- **Scalable Pre-training Approach:** We introduce a scalable framework for pretraining on diverse graphs using a Perceiver-based encoder with latent tokens, which efficiently handles graphs with varying sizes and topologies. Our approach includes advanced multi-graph sampling techniques that optimize GPU utilization, enabling large-scale pretraining across a wide range of graph datasets.

- **Demonstration of Benefits from Across-Graph Pretraining:** We show that pretraining on diverse graphs significantly improves the model's ability to generalize and transfer knowledge to unseen graphs. This demonstrates that a generalist model can leverage common structural features across different datasets to outperform specialized models.

- **Scaling Analysis and Impact of Multi-Graph Pretraining:** We provide the first scaling analysis for multi-graph pretraining on different domains, showing that larger models pretrained on more diverse graph datasets result in better generalization. Our results highlight that increasing both the scale of the model and the diversity of the training data improves performance on downstream tasks.

## 2 METHODS

In this section, we describe our method, including the model architecture and tokenization (Section 2.1.1 ), our proposed multi-task node decoder for jointly solving node classification and regression tasks by querying from the latent space (Section 2.1.2 ), and efficient tools for scaling (Section 2.2) that allowed us to build a large pretrained model that could integrate the extreme diversity in our pretraining set.

### 2.1 MODEL

#### 2.1.1 TOKENIZING DIVERSE GRAPHS

Each graph is represented as a sequence of node-level tokens, where each token embedding encodes both the node features and a positional embedding of the node. Let $\mathcal{D} = \{\mathcal{G}_g\}_{g=1}^G$ denote a dataset containing $G$ graphs, where each graph can be expressed in terms of its node and edges as

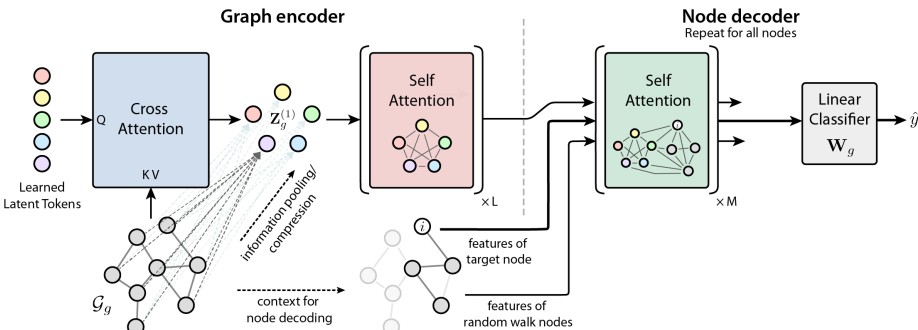

Figure 1: **Overview of GRAPHFM architecture and multi-graph training approach**: The input node-level tokens are passed through a cross-attention layer, followed by multiple self-attention layers to generate a compressed graph-level representation (latents). We decode node-level properties by creating a spatial sequence with features from a query node, a subset of its neighbors and the latents, which is then processed by a node decoder that uses self attention across the sequence.

$\mathcal{G}_g = (V_g, E_g)$, with node features $\{\mathbf{u}_i\}_{i=1}^{N_g}$. To process a graph with a transformer, we start by building a sequence of tokens as $\mathbf{X}_g = [\mathbf{x}_1, \dots, \mathbf{x}_{N_g}]$, where $\mathbf{x}_i$ concatenates a projection of the node features using a Multi Layer Perceptron (MLP), $\tilde{\mathbf{u}}_i = \mathrm{MLP}_g(\mathbf{u}_i)$, and the positional encoding (PE), $\mathbf{p}_i$, of the $i^{\text{th}}$ node. We use SignNet (Lim et al., 2022) which computes sign-invariant features from the eigenvectors of the graph Laplacian and uses this as a basis for alignment of PE tokens across all the graphs.

To build a model that can be trained across diverse graphs, we propose to tokenize each graph into a fixed and common latent space using a Perceiver encoder (Jaegle et al., 2021a). This encoder learns a set of latent query tokens which, using a cross-attention operation, query the nodes in the input graph and produce a compressed representation of it in the latent space. In the context of graphs, we can think about this as a way of routing communication between distant nodes by first going through a small number of learnable "virtual nodes" (Figure 1) that are compressed from the input graph.

For all graphs, we maintain a shared sequence of $K$ learned latent tokens $\mathbf{Z}_0 = [\mathbf{z}_{0,1}, \dots, \mathbf{z}_{0,K}]$, with $\mathbf{z}_{0,i} \in \mathbb{R}^D$ and $K$ considerably smaller than the size of most graphs, in this work $K = 512$. Node embeddings in the input graph are then compressed via a cross-attention operation:

$$\mathbf{Z}_g^{(1)} \leftarrow \text{Cross-Attn}(\mathbf{Q}_g, \mathbf{K}_g, \mathbf{V}_g) = \mathbf{Z}^{(0)} + \text{softmax}\left(\frac{\mathbf{Q}\mathbf{K}_g^T}{\sqrt{d_k}}\right)\mathbf{V}_g, \qquad (1)$$

where the queries, $\mathbf{Q} = \mathbf{W}_q\mathbf{Z}_0$, are projections of the learnable virtual node tokens, while the keys and values are projections of the graph's token embeddings: $\mathbf{K}_g = \mathbf{W}_k\mathbf{X}_g$ and $\mathbf{V}_g = \mathbf{W}_v\mathbf{X}_g$, where the key and value weight matrices are shared by all the graphs. This operation is followed by a series of $L$ self-attention blocks in the latent space to obtain a sequence of $K$ latent tokens, $\mathbf{Z}_g^{\text{out}}$. We use the standard transformer block with pre-normalization layers and feed-forward nets (Vaswani, 2017). Note that the complexity here is $KN_g + LK^2 \ll N_g^2$; When the number of latent tokens $K \ll N_g$, this results in a significant reduction in compute and memory.

*Remark.* Compressing every graph into a fixed set of virtual node embeddings, allows us to build a learnable "shared vocabulary" across graphs, and leverage common semantic and topological patterns across datasets and domains. Additionally, this approach also allows us to better integrate graphs of variable sizes, since most of the computation happens in the self-attention blocks, where all graphs are represented by an equally sized sequence of latent tokens.

### 2.1.2 NODE DECODER

Our encoder model is designed to do the bulk of the computation when processing the graph. To be able to readout node-level features, we developed a multi-task node decoder that combines the virtual node embeddings learned by our encoder $\mathbf{Z}_g^{\text{out}}$ with local information from a node and its neighbors to create a sequence $\mathbf{S}_g^i$ that can be processed by a transformer to produce a final node-level estimate of it's class information.

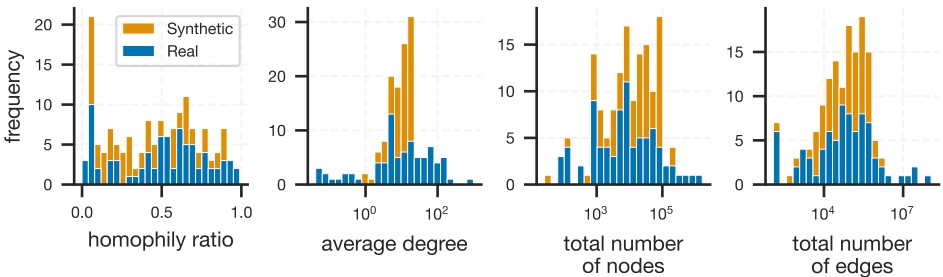

Figure 2: **Characteristics of graph datasets used to train GraphFM:** From left to right, we compute the histograms of the homophily ratio, average degree, number of nodes and number of edges of all 152 graphs used during training. The homophily ratio provides a measure of how frequently a node is directly connected to other nodes from the same class.

The sequence $\mathbf{S}_g^i$ for the $i$th node can be represented as:

$$\mathbf{S}_g^i = \Big[ \underbrace{(\mathbf{x}_i; \tau_{\text{self}})}_{\text{node}}, \underbrace{(\mathbf{x}_{\mathcal{N}_i^1}; \tau_{\text{neighbor}}) \ldots (\mathbf{x}_{\mathcal{N}_i^T}; \tau_{\text{neighbor}})}_{\text{neighbors}}, \underbrace{(\mathbf{Z}_1^{\text{out}}; \tau_{\text{latent}}) \ldots (\mathbf{Z}_K^{\text{out}}; \tau_{\text{latent}})}_{\text{virtual latent nodes}} \Big], \qquad (2)$$

where $\mathbf{x}$ and $\tau_{\text{type}}$ denote the features and their token type (latent, self, or neighbor), respectively, and $\mathcal{N}_i^j$ denotes the $j^{\text{th}}$ neighbor selected in the neighborhood of node $i$. We use a small encoder-only transformer with a depth of $M$ to obtain a final set of embeddings $\mathbf{S}_g^{\text{out}_i}$ for node $i$. Note that the complexity is $N_g M (K + T + 1)^2 \ll N_g^2$.

### 2.1.3 MULTI-TASK PRETRAINING ON A VARIETY OF NODE CLASSIFICATION AND REGRESSION TASKS

In the end, a per-dataset linear classifier (or regressor) $\mathbf{W}_g$ is tasked with producing the final predictions $\hat{y}_i$ for node $i$, mapping the final embedding of node $i$, the first token in the $\mathbf{S}_g^i$ sequence, to the output space as: $\hat{y}_i = \mathbf{W}_g^T \mathbf{S}_g^{i,\text{out}}$. The linear projection effectively translates the node-level embeddings into task-specific outputs, such as class labels for classification or continuous values for regression. The model handles a wide variety of tasks across different datasets, such as citation graphs are trained to predict academic fields and co-purchasing graphs are used to predict product categories. Each dataset has an arbitrary label space, varying not only in the number of labels but also in the nature and semantics of the output classes.

*Remark.* Since this model is trained end-to-end, the model learns how to optimally route and query information on graphs to maximize the performance on the various pre-training tasks. The virtual nodes allow for longer-range and global interactions to be encoded in the virtual node embeddings, and uses this information along with the local information provided by the node's neighbors.

## 2.2 IMPORTANT INGREDIENTS FOR TRAINING ON DIVERSE GRAPHS

### 2.2.1 MULTI-GRAPH PACKING

Typically when creating batches for training graph transformers, padding is used to extend the smaller graphs to have the same size as the largest graph in the batch (Rampášek et al., 2022; Ying et al., 2021). This approach is likely inherited from the transformer architectures found in other domains where the context window (or sequence length) is usually fixed. But for graphs, the problem with padding is particularly pronounced when there is a significant size disparity among different graphs in the same batch. Alternative solutions exist, and in particular, the graph community have been pioneers in batching variable-sized graphs. Message-passing frameworks combine multiple graphs into a single large graph over which message passing is conducted (Fey & Lenssen, 2019b; Krell et al., 2022). However, these out-of-the-box implementations are not suited for transformers which use fully-connected attention.

We implement a custom data collator, which merges all graphs in the batch into a single large sequence of tokens, and adapts the attention mask to restrict each graph to itself. In particular, we

leverage Flash Attention (Dao, 2023) which makes computing attention over very large sequences extremely efficient. By doing so, we avoid any superfluous padding, and this in turn improves the computational efficiency during training.

### 2.2.2 BALANCED GPU UTILIZATION WITH THE DISTRIBUTEDSSSAMPLER

During multi-GPU distributed training, a global batch is formed by randomly sampling graphs from different datasets, which is then equally split among the GPUs. Naively splitting the batch can lead to unbalanced GPU utilization. On one hand, we can have a large batch of relatively small graphs, and another where we can only have a batch with one or two very large graphs. This means that we would be forced to lower the batch size, to avoid going out of memory when multiple large graphs are batched together. Our *Distributed Snake Strategy Sampler* (DistributedSSSampler) employs a bidirectional filling strategy, where graphs, sorted by their size, are distributed in a snake-like pattern, initially assigned to GPUs from right to left, then left to right and so on. This method effectively pairs large graphs with small ones in subsequent passes, preventing the concentration of multiple large graphs on the same GPU, thus achieving efficient load balancing and uniform GPU utilization. A detailed algorithm and more details are provided in Appendix C.1.

We show the effectiveness of this approach in Figure 3, where we demonstrate significantly lower variance in GPU load compared to the default PyTorch batch sampler and near 100% utilization. The effectiveness is more pronounced the more GPUs are used[1]. This subsequently allows us to use substantially larger batch sizes, resulting in further improvement in stability and a significant 2 to 4x speed-up in training time.

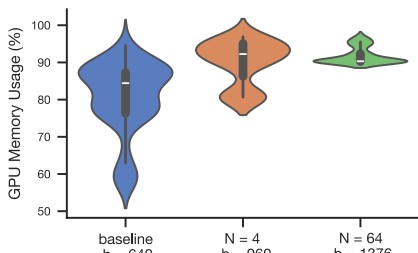

### 2.2.3 OVERALL TIME AND MEMORY SAVINGS

In total, our largest model, trained on all the pretraining data, takes ~6 days to train on 8 A40 GPUs for 300 epochs. With our distributed sampler, each epoch takes approximately 56 minutes (0.93 hours), compared to 299.04 minutes (~5 hours) without it. By using the distributed sampler, we observe a speedup of approximately 5.53x, reducing the total training time from 33 days to 6 days.

Figure 3: **The computational benefits of using our multi-graph sampling approach**: GPU memory utilization during distributed training when using the default batch sampler with 8 GPUs (left) vs. our DistributedSSSampler for N=4 (middle) and N=64 (right) GPUs. The total batch size is $N \times b$.

Please refer to Appendix C for an ablation study on the proposed sampler and multi-graph packing methods.

## 3 DATASETS

In standard practice, one would train on individual datasets, one at a time. However, to build our large multigraph model, we needed to curate a large dataset of graphs that have varied structures, features, and tasks.

**Datasets used for pretraining.** For pre-training, we curated a large set of 80 real-world graph datasets from the PyTorch Geometric library (Fey & Lenssen, 2019a) and Network Repository (Rossi & Ahmed) (Figure 2). These datasets span a wide range of domains, including: citation networks, product recommendation graphs, webpage traffic graphs, biological protein-protein interactions, and molecular graphs, and vary in their degree of heterophily (extent to which neighbors share the same class or node-level labels). Each dataset contributes unique structural patterns and tasks, providing a rich source for our model to learn diverse graph representations. In addition to these realworld datasets, we generated 72 synthetic graphs (Tsitsulin et al., 2022) that vary in their hetero- and homophily ratios and overall size and density (see Appendix B.1). We note that most datasets used in popular benchmarks were left out of pretraining in order to test the pretrained model on these datasets in out-of-distribution (OOD) finetuning.

In Figure 2, we show a summary of various graph statistics, including the number of nodes and edges, the average degree of each node, and the homophily ratio of the graph. The homophily ratio ranges

---

[1]The same effect can be obtained using gradient accumulation when resource bound. See Appendix C.1

from 0 to 1 and encodes the average amount of nodes with nearest neighbors from the same class. When comparing our realworld datasets with the synthetic graphs added to the mix (Figure 2), we see a good amount of overlap between most features except for the average degree. The average degree of realworld graphs spans a larger range, and the synthetic graphs have a more limited range. We also find an enrichment of heterophilic graphs with low homophily ratio in the added synthetic data. In total, we counted more than 7.4M nodes and 163.9M edges across all 152 datasets used for pretraining We point the reader to Appendix B.1 for a detailed description of all datasets.

**Datasets for testing out-of-distribution transfer**    To demonstrate the adaptability of our pretrained model through fine-tuning on unseen data (out-of-distribution datasets), we leverage a smaller, but equally diverse set of graph datasets that are commonly used as benchmarks (see Appendix B.4). We use 10 different datasets that range from academic collaboration networks like "Coauthor-CS" and "Coauthor-Physics" (Sinha et al., 2015) to webpage link datasets such as "Chameleon" and "Squirrel" (Rozemberczki et al., 2021), which are particularly challenging due to their low homophily ratios, indicating less connectivity within the same class. These datasets not only test the transferability of the learned representations but also highlight the model's capability in handling graphs with varied node degrees and class distributions.

## 4    RESULTS

### 4.1    EXPERIMENTAL SETUP

**Training:**    To train all of our models, we employed the LAMB optimizer (You et al., 2019) with a learning rate of $10^{-4}$. The learning rate is scheduled based on a linear warmup of 2 epochs, followed by cosine decay until the end of training. We use `bfloat16` mixed-precision and flash attention (Dao, 2023) for higher compute efficiency while training. We trained our largest model (75M parameters) for 6.4 days on 8 NVIDIA A40 GPUs. We point the reader to further details on the architecture and model training in Appendix A.1.

**Baselines:**    We compared GRAPHFM against six baseline models that were consistently reported in both heterophilic and homophilic benchmarks. This included two GNN-based models: GCN(Kipf & Welling, 2016) and GAT(Velickovic et al., 2017), two transformer-based models: SAN (Kreuzer et al., 2021) and NAGphormer (Chen et al., 2022b), and two heterophily-based models: MLP and H2GCN (Zhu et al., 2020). For all of the baseline models, we include the best reported accuracy, and when there are no reported results for a dataset, we extensively tuned each model as in standard practice (see Appendix B.4). We also provide additional baselines in Appendix D.4 reported for subsets of the datasets tested.

**Evaluation:**    To evaluate the generalization of our pretrained model on new datasets that it hasn't encountered during pretraining, we employed two fine-tuning strategies: (i) Low-resource MLP fine-tuning (MFT), where we freeze the encoder and node decoder weights and only update the feature MLP weights, and (ii) combined MLP and node decoder fine-tuning (NFT), where we also adapt the node decoder weights. MFT is aimed at evaluating near out-of-the-box performance by leveraging the model's pretrained knowledge, with minimal additional training, whereas NFT allows for more flexibility by adjusting weights of the pretrained node decoder to better align with the OOD data. For all the fine-tuning experiments, we used a learning rate of $10^{-3}$ and a weight decay of $10^{-5}$, optimized using the AdamW optimizer (Loshchilov & Hutter, 2017), and use a gradual unfreezing strategy to update the node decoder weights in our NFT experiments. Further details are provided in the Appendix A.3.

### 4.2    EXPERIMENTS

#### Q1: IS IT POSSIBLE TO BUILD A LARGE MODEL SPANNING MANY DOMAINS?

Recent efforts in graph neural networks (GNNs) have shown success in training models on many graphs (Beaini et al., 2023; Mao et al., 2024). However, these approaches primarily focus on graphs with homogeneous structures, limiting their ability to generalize across different types of graphs. In this experiment, we aim to address a more ambitious question: can we effectively train a large model on diverse, multi-graph datasets that vary significantly in their topologies, features, and downstream

classification tasks? Our goal is to determine whether a generalist model can span multiple graph domains and improve out-of-distribution (OOD) performance through diverse pretraining.

We trained three different model sizes: a small model with 389K parameters, a medium model with 18M parameters, and a large model with 75M parameters. Each model was pretrained on progressively larger datasets containing different amounts of graph data, ranging from 200K tokens (small), to 2M tokens (medium), and finally to 7.3M tokens (large), created by taking random subsets of the largest dataset (refer to Appendix B.2 for more details). The datasets span a variety of real-world graph types and structures, as described in Section 3. For the largest scale of data, we also introduced synthetic graphs into the mix to further test the model's ability to generalize across highly diverse graph structures. The synthetic graphs provided additional variability in both topologies and node features, allowing us to assess how well the model can handle graph data that extends beyond typical real-world scenarios.

To evaluate how well the pretrained models generalize to new, unseen data, we applied our lightweight MLP fine-tuning approach (MFT) on a set of nine held-out datasets. These include four homophilic datasets (Coauthor-cs, Coauthor-physics, Amazon-photos, and Amazon-comp) and five heterophilic datasets (Texas, Wisconsin, Actor, Squirrel, and Chameleon). As illustrated in Figure 4A, we observe that performance on these OOD datasets improves consistently as the data size increases. Notably, the largest model, trained on the full 7.3M tokens, achieves a 2.1% improvement in accuracy compared to the smaller models.

We further stratified our pretraining dataset to investigate the effects of cross-domain training by creating three models: (i) "Soc" with social domain graphs (1.3M tokens), (ii) "Soc + Bio" with social and biological graphs (2M tokens), and (iii) "All" with all data, including synthetic graphs (7.3M tokens). As shown in Figure 4B, adding biological datasets improved performance on both Coauthor-CS (citation domain) and Amazon-Photo (co-purchasing network). This suggests that performance continues to scale even if the additional data is from seemingly unrelated domains (refer to Appendix D.1 for additional results).

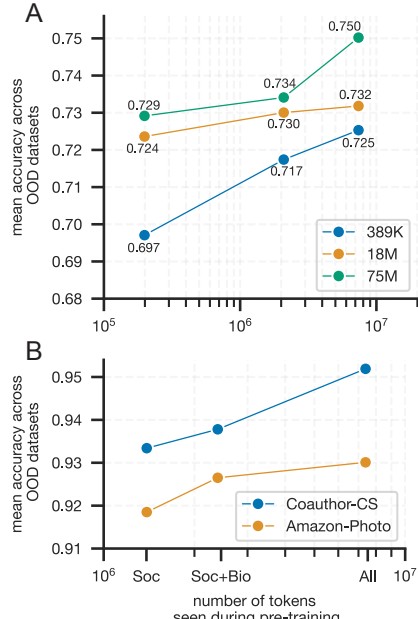

Figure 4: **Scaling Analysis**: **(A)** Average accuracy across OOD datasets (MFT) for model sizes (389K, 18M, 75M) and token counts (200K, 2M, 7.3M) seen during pre-training, using random splits of the pretraining data. **(B)** Accuracy on Coauthor-CS (citation domain) and Amazon-Photo (co-purchasing network) for the 75M model across different domain-wise pre-training splits.

These results underscore the importance of both model scale and data diversity. With more data diversity and larger models, the pretrained model demonstrates stronger generalization capabilities. This scaling analysis provides strong evidence that cross-domain pretraining enables better OOD performance, further validating the benefits of training on diverse datasets. Detailed configurations for each model size are provided in Appendix A.1.

Q2: How does our generalist approach compare with others?

Next, we compared the performance of our largest model (75M) trained on all of the data, alongside a number of message passing architectures and state-of-the-art transformer-based models. To adapt our approach to new datasets, we freeze our pretrained encoder and then finetune either the feature MLPs (MFT) or the feature MLPs and node decoder parameters (NFT). In both of these cases, we use the same hyperparameters (learning rate = $10^{-3}$) to finetune the model. In the case of NFT, we also incorporate a simple unfreezing schedule for updates which adds an additional two hyperparameters that we need to tune (refer to Appendix A.3.2).

On both homophilic and heterophilic benchmarks (Table 1), GRAPHFM performs on par with state-of-the-art specialist methods that are trained from scratch on each dataset. While the best-performing model among the baseline methods varies across datasets, GRAPHFM consistently ranks among

Table 1: **Results on a variety of homophilic and heterophilic node classification benchmarks.** From left to right, we show different message passing and graph transformer architectures, and then GRAPHFM in both the lightweight MLP-only finetuning (MFT) and node decoder finetuning (NFT). The top three numbers are bold, with the highest in bright red fading to black. Models are ranked on all 10 datasets and the average and standard deviation ranking is at the bottom.

| | | GCN | MLP | GAT | H2GCN | SAN | NAG | GraphFM-MFT | GraphFM-NFT |
|---|---|---|---|---|---|---|---|---|---|
| Homophilic | Physics | 95.38±0.20 | 95.12±0.26 | 95.14±0.28 | 96.28±0.13 | **96.83±0.18** | **96.66±0.16** | 96.64±0.17 | **96.77±0.12** |
| | CS | 94.06±0.16 | 92.99±0.51 | 93.61±0.14 | 94.02±0.31 | 94.16±0.36 | **95.00±0.14** | **95.19±0.21** | **95.24±0.18** |
| | Photo | 85.94±1.18 | 88.66±0.85 | 87.13±1.00 | 91.56±0.80 | **94.17±0.65** | **94.64±0.60** | 93.01±1.82 | **94.37±0.35** |
| | Computer | 89.47±0.46 | 84.63 | **90.78±0.13** | 89.33±0.27* | 89.83±0.16 | **91.22±0.14** | 89.95±0.83 | **90.07±0.21** |
| | Ogbn arxiv | **70.40±0.10** | 52.63±0.12 | 67.56±0.12 | 68.29±0.67 | 69.17±0.15 | 68.21±0.02* | **69.96±0.21** | **70.01 ± 0.18** |
| Heterophilic | Texas | 55.14±5.16 | **80.81±3.31** | 52.16±6.63 | **84.86±7.23** | 60.17±6.66 | 68.37±5.27* | **80.81±2.76** | 82.16±3.24 |
| | Wisconsin | 51.76±3.06 | **85.29±3.31** | 49.41±4.09 | **87.65±4.98** | 51.37±2.08 | 68.23±5.99* | 83.13±2.35 | **83.62±3.21** |
| | Actor | 27.32±1.10 | **36.63±0.70** | 27.44±0.89 | 35.70±1.00 | 27.32±1.10 | 34.33±0.94* | **36.29±0.63** | **38.01±1.07** |
| | Chameleon | 38.44±1.92 | 46.21±2.99 | 38.44±1.92 | **60.11±2.15** | 44.32±1.73* | 57.39±0.02* | **58.64±1.24** | **59.12±1.64** |
| | Squirrel | 31.52±0.71 | 28.77±1.56 | 36.77±1.68 | 36.48±1.86 | 30.92±2.14* | **49.93±0.07*** | **42.80±1.54** | **42.98±1.62** |
| **Avg Rank** | | 5.8 ± 1.9 | 6.3 ± 2.5 | 6.2 ± 1.8 | 4.0 ± 2.5 | 4.7 ± 2.2 | 3.3 ± 1.8 | 3.3 ± 0.6 | 2.1 ± 0.7 |

* This result was missing from existing literature and was obtained through extensive hyperparameter tuning.

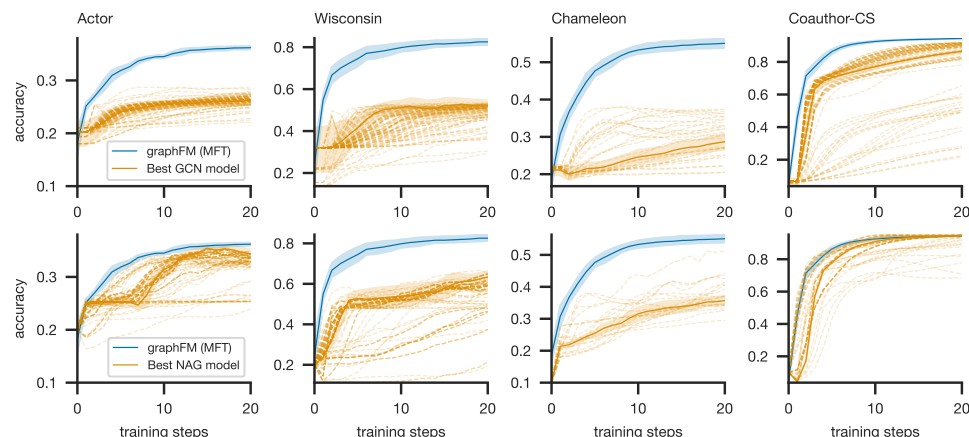

Figure 5: **Analysis of the learning dynamics.** Learning dynamics for 100 (A) random GCN and (B) NAG (NAGphormer) models compared against our lightweight finetuned model GraphFM (MFT) for four datasets. GRAPHFM works out of the box and achieves rapid learning on new datasets with few training steps, while the other approaches are less stable and often require early stopping with decreased performance over training.

the top three: GRAPHFM (NFT) achieves the highest average rank overall and GRAPHFM (MFT) is tied for the second position with NAG. Note that GRAPHFM (MFT) demonstrates significantly lower variance in rank, indicating more stable performance compared to NAG, which exhibits a more bimodal distribution in its ranking.

In contrast, baseline models may excel on a few datasets but perform poorly on others. For example, H2GCN is a top performer on heterophilic datasets but struggles with homophilic ones, whereas NAG shows the opposite behavior. These baseline models are highly specialized and designed for specific types of datasets, limiting their generalization across diverse graph types. Additionally, it's important to note that all baseline models underwent extensive hyperparameter tuning, whereas GRAPHFM performs consistently well without any further tuning. Furthermore, NFT provides significant benefit for datasets like photos and actor. By making part of the pre-trained model learnable, we are able to better adapt to the OOD datasets.

Q3: HOW DOES OUR MODEL GENERALIZE OUT-OF-THE-BOX?

A major challenge in applying graph-based models is the extensive tuning often required to achieve competitive performance. Most models are highly sensitive to hyperparameters like learning rate, depth, and weight decay. Tuning these hyperparameters, especially across datasets with different graph topologies and sizes, requires significant time and computational resources, and even then, finding a good configuration can be difficult (Guo et al., 2022; Tsitsulin et al., 2022).

In contrast, GRAPHFM offers strong out-of-the-box performance without requiring any significant tuning. To demonstrate this, we evaluated GRAPHFM using the same fixed learning rate and weight decay across multiple datasets (learning rate = $10^{-3}$, weight decay = $10^{-5}$) and observed stable and high performance across all datasets (Figure 5). Fine-tuning GRAPHFM with our simple MFT strategy resulted in low variance and rapid convergence, without the need for extensive hyperparameter exploration. This makes GRAPHFM highly efficient and cost-effective compared to models that require substantial tuning.

To highlight this contrast, we compared the performance of GRAPHFM with 100 randomly configured versions of GCN and the best performing transformer-based NAGformer (Chen et al., 2022b). Both baseline models exhibit a wide range of performance depending on the hyperparameter choices, with some configurations leading to significant instability or poor results. For instance, in the Texas dataset, GCN required exhaustive exploration of hyperparameter settings to find a stable and effective configuration. Similarly, NAGformer's performance fluctuated greatly depending on the dataset and the selected parameters, further emphasizing the cost of tuning.

Additionally, GRAPHFM demonstrates quick convergence, reaching near-optimal performance within 10-20 training steps, in stark contrast to GCN, which required considerably more iterations to converge. This efficiency is a direct result of leveraging a pretrained model, which allows GRAPHFM to start from a robust initialization and quickly adapt to the target task. The reduced need for hyperparameter tuning and faster convergence further solidify the advantages of pretraining in minimizing computational overhead and time-to-solution. Ultimately, our results position GRAPHFM as a cost-effective and reliable choice for a wide range of node classification tasks.

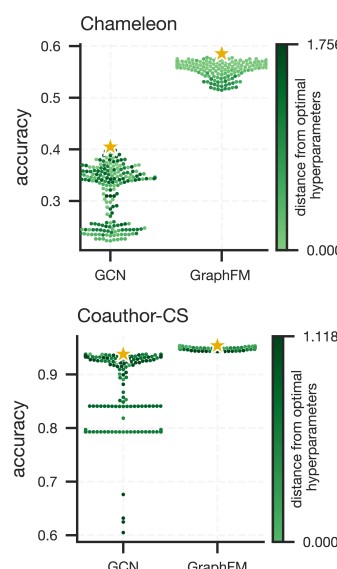

Figure 6: **Comparison of model sensitivity.** The performance of GCN and GRAPHFM for 100 different random hyperparameters on Chameleon and Coauthor-CS. Star denotes the model with the optimal hyperparameters, and the color indicates the $\ell_2$-distance between the optimal solution and each model's hyperparameters.

Q4: How stable are the solutions?

Graph-based models are highly sensitive to hyperparameter configurations, where even small deviations from optimal settings can lead to substantial performance degradation. This sensitivity poses significant challenges for ensuring stable and robust deployment. Thus, we wanted to examine the stability of model performance by exploring the performance landscape around the optimal hyperparameter configuration. We analyze the performance of both a GCN and GRAPHFM (MFT) on Coauthor-CS (homophilic) and Chameleon (heterophilic) datasets for different hyperparameters around the optimal solution (Figure 6). The set of hyperparameters is marked with a star, and other models are colored based on the normalized $\ell_2$-distance of their hyperparameter vectors to the optimal hyperparameter vector. For GRAPHFM, we observe that the distribution is concentrated around the optimal point, suggesting low sensitivity to the choice of the hyperparameters used for finetuning. We also observe that the relationship between hyperparameter deviation and performance is linear. On the other hand, for the GCN model, small deviations in hyperparameters can lead to large changes in performance, suggesting instability of the model with respect to the hyperparameters and a much noisier landscape around the optimal model.

## 5 RELATED WORK

**Graph foundation modeling approaches.** Foundation models have achieved significant success for language, vision and timeseries data (Radford et al., 2018; Dehghani et al., 2023; Das et al., 2023). These models are pre-trained on large datasets and can be adapted to a wide range of downstream tasks, effectively utilizing both prior knowledge from the pre-training stage and data from the downstream tasks to enhance performance (Brown et al., 2020). The concept of foundation models has recently extended into graph learning, leading to the development of Graph Foundation Models (GFMs) (Ibarz et al., 2022; Beaini et al., 2023; Galkin et al., 2023; Mao et al., 2024). These

models aim to generalize across diverse graph-structured data by leveraging large-scale pretraining, similar to foundation models in vision and language domains. Recent efforts have primarily focused on domain-specific GFMs, such as Mole-BERT for molecular graphs, which utilizes pretraining to improve property prediction for molecules and materials (Xia et al., 2023). Additionally, large-scale models like MatterSim (Yang et al., 2024), designed to predict molecular behaviors across different elements and conditions.

Beyond domain-specific applications, GFMs are increasingly being developed for more general tasks. Similarly, recent advancements have explored scaling laws in graph models, showing that larger models can lead to improved transfer learning and generalization (Liu et al., 2024). Similar to theirs, our work shows that scale improves performance. However, unique from all of these works is our result for cross-domain pretraining to enhance generalization across diverse graph topologies. Triplet-GMPNN (Ibarz et al., 2022) which is a foundational GNN for algorithmic reasoning tasks that is trained to perform various tasks from the CLRS benchmark (Veličković et al., 2022), or ULTRA (Galkin et al., 2023), a foundation model for knowledge graphs trained on graphs with arbitrary entity and relation vocabularies. Recent work has also shown how to use language modeling to help unify many graphs (Liu et al., 2023b).

**Scaling up graph transformers.** Graph transformers bypass standard local learning rules in GNNs by allowing all nodes on the graph to interact through self-attention (Dwivedi & Bresson, 2020). However, due to the high computational cost and benefits of the inductive bias in message passing, a number of methods have been proposed to move beyond full self-attention or combine transformers with GNNs. One class of methods combine transformer blocks with GNNs, including GraphTrans (Wu et al., 2021), GraphGPS (Rampášek et al., 2022), and SAT (Chen et al., 2022a). Another strategy is to reduce the computational complexity by using the transformer module on a coarsened or compressed graph. For instance, ANS-GT (Zhang et al., 2022) introduced a node-sampling-based graph transformers, incorporating hierarchical attention and graph coarsening, and Gapformer (Liu et al., 2023a) uses k-hops local pooling and global pooling to coarsen the large graph into a smaller set of nodes. Exphormer (Shirzad et al., 2023) coarsens the graph by doing computation through expander graphs (Deac et al., 2022). This idea of compression has also been studied through the lens of "skeletonization" (Cao et al., 2024) where they learn to identify uninformative background nodes (Cao et al., 2024) and use this information to compress them to achieve competitive performance with as little as 1% of the nodes in the graph. Many of these approaches leverage virtual nodes to facilitate message passing across large graphs, however, the compression techniques used in these works are often based on heuristics like pooling layers or expander graphs, in contrast to our work where the compression is fully learned.

## 6 CONCLUSION

In this paper, we introduced GRAPHFM, a novel approach for multi-graph pretraining that effectively handles diverse graph datasets across various domains. A key finding of our work is the positive effect of scaling both model size and data diversity. We show that cross-domain pretraining leads to better out-of-distribution performance, proving that the inclusion of diverse graph types significantly enhances the model's ability to adapt to new, unseen data. This reveals the potential for graph foundation models to benefit from combining datasets across domains, facilitating more efficient and robust training processes.

While our results are promising, there are several areas for future exploration. Our current work primarily focuses on node-level classification tasks; extending GRAPHFM to support tasks like graph-level classification, link prediction, and self-supervised learning could broaden its applicability. Moreover, expanding the diversity of pretraining datasets, such as including point clouds, mesh graphs, or knowledge graphs, may further enhance the model's generalization capabilities and impact across domains.

Looking ahead, we believe that generalist graph models like GRAPHFM have the potential to transform a variety of fields, particularly in scenarios where data is scarce or incomplete. Our work represents an important step toward more universal and adaptable graph models, and we anticipate further research into cross-domain pretraining as a promising direction for the future of graph learning.

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

APPENDIX

# A    ADDITIONAL MODEL DETAILS

## A.1    MODEL CONFIGURATION DETAILS

We used pretrained 3 configuration of models—small (398K), medium (18M) and large (75M)—for our analysis. Details of the configuration for each model are given in Table A1. In the first cross-attention layer, we used flash attention, whereas for all subsequent attention layers, we used memory-efficient attention. Both implementations were sourced from the xFormers library (Lefaudeux et al., 2022).

Table A1: Architectural details of GraphFM for different parameter sizes used in Section 2.2

| Parameter Count | 75M | 18M | 389K |
|---|---|---|---|
| **Num Latents** ($K$) | 512 | 256 | 32 |
| **Latent Dimension** | 512 | 256 | 32 |
| **Cross-Attention** | | | |
| Heads | 4 | 4 | 4 |
| FFN hidden dim | 2048 | 1024 | 128 |
| **Self-Attention** | | | |
| Depth ($L$) | 12 | 10 | 4 |
| Heads | 8 | 4 | 4 |
| FFN hidden dim | 2048 | 1024 | 128 |
| **Node Decoder** | | | |
| Depth ($M$) | 4 | 4 | 2 |
| Heads | 8 | 4 | 4 |
| FFN hidden dim | 2048 | 1024 | 128 |

## A.2    RESCALING THE LEARNING RATES FOR DIFFERENT GRAPH SIZES

When training on variable sized graphs, the MLP and linear decoder for each dataset receive updates based on the number of nodes from their respective datasets present in the batch and thus smaller graphs get updated less frequently when compared to large graphs. To mitigate this imbalance, we implemented dataset-specific learning rates for the feature MLP and linear decoders. Since they receive updates less frequently, when they do, we use a larger learning rate to update them. Without this adjustment, the weights of the common Perceiver encoder and node decoder would advance more quickly than those of the dataset-specific components, potentially leading to suboptimal learning for smaller datasets.

## A.3    FINE-TUNING STRATEGIES

In our evaluation of GraphFM's generalization capability, we employed two fine-tuning strategies aimed at adapting the model to out-of-distribution (OOD) datasets.

### A.3.1    LOW-RESOURCE MLP FINE-TUNING (MFT)

This approach is designed to assess how well the pretrained model performs out-of-the-box on different OOD graphs without extensive training. In MFT, we freeze the pretrained model and only fine-tune a lightweight multi-layer perceptron (MLP) on top of the learned representations. This

strategy allows us to quickly adapt the model to new tasks while retaining the majority of the original learned parameters. MFT is particularly useful in low-resource settings, where computational power or time is limited, as it requires minimal additional training while still providing insight into how well the pretrained model generalizes. For all MFT experiments, we used a learning rate of $10^{-3}$ and a weight decay of $10^{-5}$, optimized using the AdamW optimizer (Loshchilov & Hutter, 2017).

### A.3.2 MLP AND NODE DECODER FINE-TUNING (NFT)

In contrast to MFT, the NFT strategy involves fine-tuning part of the and is recommended when sufficient computational resources are available and the goal is to extract the maximum performance from the model. In NFT, we gradually unfreeze the node decoder, enabling the model to more effectively adapt to the new dataset. Specifically, we set a predefined epoch $U$ at which unfreezing begins, starting from the bottom layers of the node decoder. After every $S$ epochs, additional layers are unfrozen in a bottom-up manner, facilitating gradual transition to full finetuning of the model. Concurrently, the learning rate is decayed by a factor of 1.5 each time a new layer is unfrozen, ensuring controlled parameter updates. For all datasets, we tune the hyperparameters $U$ and $S$, with $U$ set to 10, 20, or 30 epochs and $S$ set to 5 or 10 epochs. This gradual unfreezing mitigates training instability, as smaller perturbations are made to higher-level feature representations. As a result, NFT allows for better adaptation, particularly for out-of-distribution (OOD) datasets, and is well suited for case when exploiting the capacity of pretrained models is critical.

## B ADDITIONAL DETAILS ON DATASETS

### B.1 PRETRAINING DATASETS

The largest model (75M parameters) was trained on 80 real world and 72 synthetic datasets. The real world datasets and their characteristics are given in Table A3.

The synthetic datasets were created using the GraphWorld (Palowitch et al., 2022) using the Stochastic Block Model (Holland et al., 1983). The generator parameters are listed in Table A2. In the graph generation process, the *node homophily ratio* is varied. The homophily is given by the following formula:

$$\frac{1}{|\mathcal{V}|} \sum_{v \in \mathcal{V}} \frac{|\{(v, w) : w \in \mathcal{N}(v) \wedge y_v = y_w\}|}{|\mathcal{N}(v)|},$$

where $\mathcal{V}$ denotes the set of all nodes in the graph, $\mathcal{N}(v)$ denotes all the neighbors of an arbitrary node $v$, and $y_v$ denotes the class membership of the node $v \in \mathcal{V}$. We classify datasets into *homophilic datasets* and *heterophilic datasets* based on the homophily score: datasets with homophily $\geq 0.5$ are classified as *homophilic datasets* and *heterophilic datasets* otherwise.

### B.2 DETAILS ON SMALL AND MEDIUM SCALE DATASET

The small and medium scale datasets, as discussed in Section 4.2, were created by taking a random subset of the large dataset(80 real and 72 synthetic).

**Dataset subset for small scale data:**   The following datasets were used to train models with small scale data: Wiki, BlogCatalog, Roman-empire, Minesweeper, Tolokers, Questions, Twitch-EN, Twitch-FR, Twitch-PT, Twitch-RU, DeezerEurope, GitHub, LastFMAsia, Airports-USA, Airports-Europe, PolBlogs and EmailEUCore

**Dataset subset for medium scale data:**   The following datasets were used to train models with medium scale data: Wiki, BlogCatalog, Roman-empire, Minesweeper, Tolokers, Questions, Twitch-EN, Twitch-FR, Twitch-PT, Twitch-RU, DeezerEurope, GitHub, LastFMAsia, Airports-USA, Airports-Europe, PolBlogs and EmailEUCore, Reddit, Reddit2, Flickr, Yelp, PPI, Facebook, Amazon-ratings, Minesweeper, Twitch-DE, Twitch-ES, FacebookPagePage, Airports-Brazil, penn94, reed98, amherst41, johnshopkins55, genius, CitationFull-CiteSeer, CitationFull-Cora-ML and CitationFull-PubMed

Table A2: Graphworld generator parameters for synthetic graphs

| Parameter Name | Description | Values |
|---|---|---|
| nvertex | Number of vertices in the graph. | [32, 500000] |
| $p/q$ ratio | The ratio of in-cluster edge probability to out-cluster edge probability. | [0.1, 10.0] |
| avg. degree | The average expected degrees of the nodes. | [1.0, 20.0] |
| feature center distance | The variance of feature cluster centers, generated from a multivariate Normal. | [0.0, 5.0] |
| num clusters | The number of unique node labels. | [2, 6] |
| cluster size slope | The slope of cluster sizes when index-ordered by size. | [0.0, 0.5] |
| power exponent | The value of the power law exponent used to generate expected node degrees. | [0.5, 1.0] |

### B.3 DETAILS ON SOCIAL AND BIOLOGY DOMAIN DATASETS

The social and biology datasets, as discussed in Section D.1 and Section 4.2, included the following subsets:

**Dataset subset for social domain:** The following datasets were used to train the social-specific model: fb-CMU-Carnegie49, Yelp,Wiki, BlogCatalog, Facebook, Twitch-DE, Twitch-EN, Twitch-ES, Twitch-FR, Twitch-PT, Twitch-RU, DeezerEurope, GitHub, FacebookPagePage, LastFMAsia, penn94, reed98, amherst41, johnshopkins55, genius and soc-pokec.

**Dataset subset for biology domain:** The following datasets were added as part of the biology domain to train the combined social and biology model: BZR, DD, DD199, DD21, DD242, DD244, DD349, DD497, DD6, DD68, DD687, DHFR, ENZYMES, ENZYMES118, ENZYMES123, ENZYMES295, ENZYMES296, ENZYMES297, ENZYMES8, KKI, OHSU, PROTEINS-full, Peking-1, Tox21_p53, gene, proteins-all and PPI.

### B.4 FINETUNING DATASETS

For our evaluations, we held out a number of datasets that are used for standard benchmarks in both larger scale node classification and heterophilic graphs.

#### B.4.1 HOMOPHILIC DATASETS

We use five real-world datasets, Amazon Computers and Amazon Photos (McAuley et al., 2015), Coauthor CS and Coauthor Physics (Sinha et al., 2015) and Obgn-Arxiv (Hu et al., 2020). Key statistics for the different datasets are listed in Table A3 in the finetuning-section. The experimental setup follows that of (Luo et al., 2022), where we split the dataset into development and test sets. All the hyperparameter tuning is done on the development set and the best models are evaluated on the test set. The runs are averaged over 20 random splits to minimize noise. We follow a 60:20:20% train/val/test split for the Amazon and Coauthor datasets. For Obgn-Arxiv we follow the experimental setup used in (Hu et al., 2020). The results for the Coauthor-Physics, Coauthor-CS, and Amazon-Photos obtained from in Table A6 have been sourced from (Liu et al., 2023a). The results for the Amazon-Comp dataset are taken from (Hoang et al., 2023) except for MLP which was obtained from (Luo et al., 2022).

Table A3: Pre-Training Datasets and their characteristics

| Dataset | Number of Graphs | Nodes | Edges | Homophily Ratio | Average Degree | Node Features | Node Classes | Learning Rate |
|---|---|---|---|---|---|---|---|---|
| BA-1_10_60-L5 | 1 | 804 | 46410 | 0.2 | 115.45 | 1 | 5 | 0.0014 |
| BA-2_24_60-L2 | 1 | 10693 | 639750 | 0.5 | 119.66 | 1 | 2 | 0.0087 |
| BZR | 405 | 35.75 | 76.71 | 0.42 | 0.07 | 1 | 53 | 0.0082 |
| CL-100K-1d8-L9 | 1 | 92482 | 373989 | 0.11 | 8.09 | 1 | 9 | 0.00064 |
| CL-10K-1d8-L5 | 1 | 10000 | 44896 | 0.2 | 8.98 | 1 | 5 | 0.00096 |
| DD | 1178 | 284.32 | 1431.32 | 0.07 | 0.058 | 1 | 89 | 0.00085 |
| DD199 | 1 | 841 | 1902 | 0.067 | 4.52 | 1 | 20 | 0.00085 |
| DD21 | 1 | 5748 | 14267 | 0.07 | 4.96 | 1 | 40 | 0.00085 |
| DD242 | 1 | 1284 | 3303 | 0.08 | 5.14 | 1 | 20 | 0.00042 |
| DD244 | 1 | 291 | 822 | 0.074 | 5.65 | 1 | 20 | 0.00085 |
| DD349 | 1 | 897 | 2087 | 0.05 | 4.65 | 1 | 20 | 0.00085 |
| DD497 | 1 | 903 | 2453 | 0.06 | 5.43 | 1 | 20 | 0.0028 |
| DD6 | 1 | 4152 | 10320 | 0.07 | 4.97 | 1 | 20 | 0.00085 |
| DD68 | 1 | 775 | 2093 | 0.072 | 5.4 | 1 | 20 | 0.0028 |
| DD687 | 1 | 725 | 2600 | 0.06 | 7.17 | 1 | 20 | 0.0028 |
| DHFR | 756 | 42.43 | 89.09 | 0.32 | 0.04 | 3 | 53 | 0.0018 |
| ENZYMES | 600 | 32.63 | 124.27 | 0.67 | 0.09 | 18 | 3 | 0.0020 |
| ENZYMES118 | 1 | 96 | 121 | 0.58 | 2.52 | 1 | 2 | 0.00087 |
| ENZYMES123 | 1 | 90 | 127 | 0.52 | 2.82 | 1 | 2 | 0.0076 |
| ENZYMES295 | 1 | 124 | 139 | 0.71 | 2.24 | 1 | 2 | 0.0076 |
| ENZYMES296 | 1 | 126 | 141 | 0.72 | 2.24 | 1 | 2 | 0.00087 |
| ENZYMES297 | 1 | 122 | 149 | 0.65 | 2.44 | 1 | 2 | 0.0020 |
| ENZYMES8 | 1 | 88 | 133 | 0.77 | 3.02 | 1 | 2 | 0.0076 |
| ER-AvgDeg10-100K-L2 | 1 | 99997 | 499332 | 0.50 | 9.99 | 2 | 2 | 0.0049 |
| ER-AvgDeg10-100K-L5 | 1 | 99997 | 499332 | 0.20 | 9.99 | 1 | 5 | 0.0013 |
| KKI | 83 | 26.96 | 96.84 | 0 | 0.39 | 1 | 189 | 0.0012 |
| MSRC-21 | 563 | 77.52 | 396.65 | 0.74 | 0.13 | 1 | 24 | 0.0063 |
| MSRC-21C | 209 | 40.28 | 193.20 | 0.61 | 0.27 | 1 | 22 | 0.0017 |
| MSRC-9 | 221 | 40.58 | 193.21 | 0.69 | 0.26 | 1 | 10 | 0.009 |
| OHSU | 79 | 82.01 | 399.32 | 0 | 0.56 | 1 | 189 | 0.0095 |
| PLC-40-30-L5 | 1 | 11025 | 437979 | 0.2 | 79.45 | 1 | 5 | 0.0086 |
| PLC-60-30-L2 | 1 | 117572 | 7045181 | 0.5 | 119.84 | 1 | 2 | 0.0013 |
| PROTEINS-full | 1113 | 39.06 | 145.63 | 0.97 | 0.05 | 2 | 8 | 0.0063 |
| Peking-1 | 85 | 39.31 | 154.71 | 0 | 0.44 | 1 | 189 | 0.0027 |
| SW-10000-6-0d3-L2 | 1 | 10000 | 30000 | 0.5 | 6 | 1 | 2 | 0.00096 |
| SW-10000-6-0d3-L5 | 1 | 10000 | 30000 | 0.2 | 6 | 1 | 5 | 0.0088 |
| SYNTHETIC | 300 | 100 | 392 | 0.18 | 0.16 | 1 | 8 | 0.0018 |
| TerroristRel | 1 | 881 | 8592 | 0.92 | 19.51 | 1 | 2 | 0.0033 |
| Tox21_p53 | 1 | 153563 | 314046 | 0.62 | 4.09 | 1 | 46 | 0.00054 |
| fb-CMU-Carnegie49 | 1 | 6637 | 249967 | 0.5 | 75.33 | 1 | 3 | 0.0010 |
| gene | 1 | 1103 | 1672 | 0.4 | 3.03 | 1 | 2 | 0.012 |
| proteins-all | 1 | 43471 | 162088 | 0.66 | 7.46 | 1 | 3 | 0.00075 |
| reality-call | 1 | 27058 | 51200 | 0.9 | 15 | 1 | 2 | 0.0071 |
| Reddit | 1 | 232965 | 114615892 | 0.76 | 983.98 | 602 | 41 | 0.0035 |
| Reddit2 | 1 | 232965 | 23213838 | 0.78 | 199.29 | 602 | 41 | 0.0035 |
| Flickr | 1 | 89250 | 899756 | 0.31 | 20.16 | 500 | 7 | 0.0051 |
| Yelp | 1 | 716847 | 13954819 | - | 38.93 | 300 | 100[1] | 0.00031 |
| Wiki | 1 | 2405 | 17981 | 0.71 | 14.95 | 4973 | 17 | 0.0012 |
| BlogCatalog | 1 | 5196 | 17981 | 0.40 | 132.21 | 8189 | 6 | 0.0099 |
| PPI | 1 | 56944 | 1612348 | 0.63 | 56.63 | 50 | 121 | 0.0016 |
| Facebook | 1 | 4039 | 88234 | 0.99 | 43.69 | 1283 | 193 | 0.0011 |
| Roman-empire | 1 | 22662 | 65854 | 0.05 | 5.81 | 300 | 18 | 0.0074 |
| Amazon-ratings | 1 | 24492 | 186100 | 0.38 | 15.2 | 300 | 5 | 0.00082 |
| Minesweeper | 1 | 10000 | 78804 | 0.68 | 15.76 | 7 | 2 | 0.0088 |
| Tolokers | 1 | 11758 | 1038000 | 0.59 | 176.56 | 10 | 2 | 0.0022 |
| Questions | 1 | 48921 | 307080 | 0.84 | 12.55 | 301 | 2 | 0.0061 |
| Twitch-DE | 1 | 9498 | 315774 | 0.64 | 66.49 | 128 | 2 | 0.0023 |
| Twitch-EN | 1 | 7126 | 77774 | 0.59 | 21.82 | 128 | 2 | 0.0010 |
| Twitch-ES | 1 | 4648 | 123412 | 0.59 | 53.10 | 128 | 2 | 0.0011 |
| Twitch-FR | 1 | 6551 | 231883 | 0.54 | 70.79 | 128 | 2 | 0.0010 |
| Twitch-PT | 1 | 1912 | 64510 | 0.58 | 67.47 | 128 | 2 | 0.0012 |
| Twitch-RU | 1 | 4385 | 78993 | 0.63 | 36.02 | 128 | 2 | 0.0011 |
| DeezerEurope | 1 | 28281 | 185504 | 0.52 | 13.11 | 128 | 2 | 0.0070 |
| GitHub | 1 | 37700 | 578006 | 0.84 | 30.66 | 128 | 2 | 0.0065 |
| FacebookPagePage | 1 | 22470 | 342004 | 0.88 | 30.44 | 128 | 2 | 0.00085 |
| LastFMAsia | 1 | 7624 | 55612 | 0.87 | 14.59 | 128 | 18 | 0.0092 |
| Airports-Brazil | 1 | 131 | 1074 | 0.46 | 16.39 | 131 | 4 | 0.0013 |
| Airports-Europe | 1 | 399 | 5995 | 0.40 | 30.05 | 399 | 4 | 0.0015 |
| Airports-USA | 1 | 1190 | 13599 | 0.69 | 22.85 | 1190 | 4 | 0.0092 |
| PolBlogs | 1 | 1490 | 19025 | 0.91 | 25.54 | 1 | 2 | 0.0013 |
| EmailEUCore | 1 | 1005 | 25571 | 0.36 | 50.89 | 1 | 42 | 0.0032 |
| penn94 | 1 | 41554 | 2724458 | 0.51 | 131.11 | 4814 | 2 | 0.0064 |
| reed98 | 1 | 962 | 37624 | 0.52 | 78.22 | 745 | 2 | 0.0032 |
| amherst41 | 1 | 2235 | 181908 | 0.53 | 162.78 | 1193 | 2 | 0.011 |
| johnshopkins55 | 1 | 5180 | 373172 | 0.55 | 144.08 | 2406 | 2 | 0.0025 |
| genius | 1 | 421961 | 984979 | 0.62 | 4.67 | 12 | 2 | 0.00040 |
| CitationFull-CiteSeer | 1 | 4230 | 10674 | 0.95 | 5.04 | 602 | 6 | 0.0011 |
| CitationFull-Cora-ML | 1 | 2995 | 16316 | 0.78 | 10.89 | 2879 | 7 | 0.0028 |
| CitationFull-PubMed | 1 | 19717 | 88648 | 0.80 | 8.99 | 500 | 3 | 0.00087 |
| soc-pokec | 1 | 1632803 | 30622564 | 0.44 | 37.51 | 500 | 3 | 0.00019 |

[1] Multi label binary classification.

Table A4: Fine-Tuning Datasets and Their Characteristics

| Dataset | Number of Graphs | Nodes | Edges | Homophily Ratio | Average Degree | Node Features | Node Classes |
|---|---|---|---|---|---|---|---|
| Actor | 1 | 7600 | 30019 | 0.21 | 7.89 | 932 | 5 |
| Amazon-Computers | 1 | 13752 | 4491722 | 0.77 | 71.51 | 767 | 10 |
| Amazon-Photo | 1 | 7650 | 238162 | 0.82 | 62.26 | 745 | 8 |
| Coauthor-CS | 1 | 18333 | 163788 | 0.80 | 17.86 | 6805 | 15 |
| Coauthor-Physics | 1 | 34493 | 495924 | 0.93 | 28.75 | 8415 | 5 |
| Chameleon | 1 | 2277 | 36101 | 0.23 | 31.70 | 2325 | 5 |

### B.4.2 HETEROPHILIC DATASETS

We use five real-world datasets with graphs that have a homophily level $\leq 0.30$, Texas, Wisconsin and Actor (Pei et al., 2020) and Chameleon and Squirrel (Rozemberczki et al., 2021). Key statistics for the different datasets are listed in Table A3 in the finetuning-section. We follow the experimental setup in (Pei et al., 2020), and use the same 10 train/val/test splits that are provided. The results for GCN based methods and heterophily based methods in Table A7 have been taken from (Azabou et al., 2023), and the results for transformer based methods have been taken from (Liu et al., 2023a)

### B.5 STANDARD HYPERPARAMETER SEARCH GRID FOR BASELINES

The hyperparameter search space grid used for tuning baselines for Table 1 is detailed in Table A5.

Table A5: Hyperparameter Search Space

| Hyperparameter | Type | Range |
|---|---|---|
| Hidden Dim | Categorical | $\{16, 32, 64, 128\}$ |
| Depth | Categorical | $\{1, 2\}$ |
| Dropout | Uniform | $[0.0, 0.9]$ |
| Learning Rate | Log uniform | $[5e\text{-}5, 5e\text{-}1]$ |
| Weight Decay | Log uniform | $[1e\text{-}5, 1e\text{-}2]$ |

## C ADDITIONAL DETAILS ON MULTI-GRAPH TRAINING

One key aspect of our work is testing scale. Thus, to build a model across large amounts of diverse graph data, we developed a number of approaches for efficient training and multi-GPU usage.

Figure A1 shows an ablation study the epoch time for various GPU optimizations we have proposed in Section 2.2. The epoch time was calculated using the medium-sized model with 18M parameters, as detailed in Appendix A.1.

**Note:** Removing chaining made it impossible to run the largest model (75M parameters) with our available computational resources (8 A40 GPUs). Therefore, we performed the ablation using the medium-sized model. This highlights the significance of our optimization techniques, which enabled us to scale up and run such large models efficiently.

### C.1 DISTRIBUTEDSSSAMPLER

In designing this sampler, we prioritized ensuring that it neither introduces bias into the data sampling process nor alters the distribution of the graphs from the datasets. Its primary function is to enhance batch construction and distribution across GPUs.

First, the sampler defines a set of $N$ buckets with a fixed node budget $B$, where $N$ can be the number of GPUs and $B$ is the node-level batch size. The graphs (across all GPUs) are sorted in descending order based upon their size. The sampler then employs a bidirectional filling strategy within the buckets. The distribution process, as described in Algorithm 1 involves distributing graphs in a snake-like pattern, initially filling from right to left, then switching to left to right and so on. When a graph is added to a bucket, it uses up part of the budget, equal to its size. This method effectively pairs larger graphs with smaller ones in

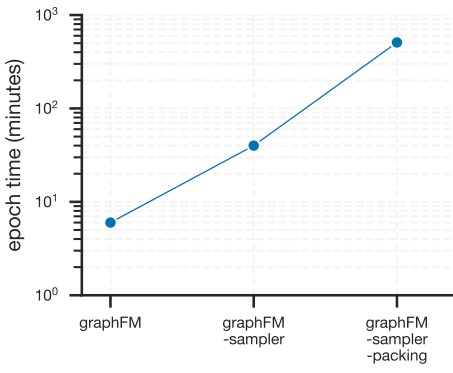

Figure A1: **Ablation for GPU optimizations**: Epoch time in minutes on removing various gpu optimizations proposed for GRAPHFM

subsequent passes, preventing the concentration of multiple large graphs on the same GPU, thus achieving efficient load balancing and uniform GPU utilization. Figure A2A shows an overview of how the sampler distributes the graphs into buckets. We find that stability is improved with a larger number of buckets $N$ (Figure A2B). When the number of GPUs is fixed, we can achieve a larger $N$ by using gradient accumulation, which artificially increases the number of buckets by a factor equal to the number of accumulation steps, without biasing the sampling process.

---

**Algorithm 1** Distribute graph nodes into virtual GPU buckets

---

1: **input:** Batch size $B$, Bucket count $N$, Graphs in the dataset $\mathcal{G} = \{\mathcal{G}_0, \mathcal{G}_1, \ldots\}$, Subgraphs sampled for this minibatch $\mathcal{G}^m = \{\mathcal{G}_0^m, \mathcal{G}_1^m, \ldots\}$
2: **precondition:** $\sum_i |\mathcal{G}_i^m| == N \times B$
3: **initialize:**
4:    $buckets \leftarrow$ array of $N$ empty arrays        # will store subgraphs in each bucket
5:    $counts \leftarrow$ array of $N$ zeroes           # will store number of nodes in each bucket
6:    $b \leftarrow 0$                                   # bucket index
7:    $d \leftarrow 1$                                     # direction
8:    Sort $\mathcal{G}^m$ according to node-counts in $\mathcal{G}$, largest graph goes first
9: **for all** $\mathcal{G}_i^m$ in $\mathcal{G}^m$ **do**
10:    **while** $|\mathcal{G}_i^m| > 0$ **do**
11:       **if** $counts[b] < B$ **then**
12:          # insert a part of $\mathcal{G}_i^m$ into bucket $b$
13:          $n \leftarrow \min(|\mathcal{G}_i^m|, B - counts[b])$
14:          $counts[b] \leftarrow counts[b] + n$
15:          append first $n$ nodes of $\mathcal{G}_i^m$ to $buckets[b]$
16:          remove first $n$ nodes from $\mathcal{G}_i^m$
17:       **end if**
18:       # go to the next bucket, switching direction at the boundaries
19:       $b \leftarrow b + d$
20:       **if** $b \geq N$ or $b < 0$ **then**
21:          $d \leftarrow -d$
22:          $b \leftarrow b + d$
23:       **end if**
24:    **end while**
25: **end for**
26: **return** $buckets$

---

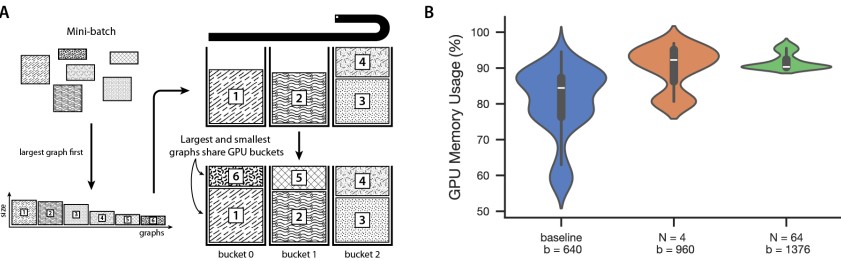

Figure A2: **Multi-GPU utilization**: **A**: A diagram visualizing our sample distribution strategy. **B**: GPU memory utilization during distributed training when using the default batch sampler vs. our DistributedSSSampler for N=4 and N=64 buckets.

## C.2 GRAPHSAINT RANDOM WALK SAMPLER

Efficient neighborhood sampling for large graphs is crucial for our node decoder, as traditional methods for k-hop neighborhood sampler often become computationally prohibitive with the increasing size and complexity of the graph data. To overcome these limitations, we have adopted the GraphSAINT Random Walk Sampler (Zeng et al., 2019), specifically designed for efficient sampling in large-scale graphs.

### C.3 RAM Optimization in Multi-GPU Environments

In multi-GPU training environments, efficient use of system memory is crucial, especially when handling large graph datasets. Traditional approaches lead to substantial memory redundancy, as each GPU process typically loads a complete dataset into system RAM. This results in each process duplicating the dataset in system memory, leading to inefficient memory usage and potential system overload.

To address this, we utilize a shared memory management approach using Python's `multiprocessing.Manager()` to coordinate dataset access across multiple GPU processes. This method ensures that each dataset is loaded into RAM only once, regardless of the number of GPUs, thereby avoiding duplication and conserving memory resources.

## D ADDITIONAL EXPERIMENTS

### D.1 Separating pretraining datasets into different domains

We further stratified our pretraining dataset to invetigate the effects of cross-domain training, and created three models that contained: (i) graph datasets from "social domains" including product graphs and citation networks (1.3M tokens), (ii) both the social datasets and all biological graphs in the dataset (Bio+Soc, 2M tokens), and (iii) compare with our model trained on all data including sytnthetic graphs (7.3M tokens).

When comparing graph features across social and biological domains, we found distinct structural differences: biological datasets generally exhibited higher levels of heterophily, lower average degree, and fewer edges, whereas social graphs showed more homophily, higher degrees, and denser connections (Figure A4B). Synthetic graphs added a wide range of characteristics, particularly increasing the number of heterophilic graphs used in pretraining, which contributed to a broader diversity of features (Figure A4A).

All three models were then fine-tuned on four homophilic datasets (coauthor-CS, coauthor-physics, amazon-photos, and amazon-computers) and five heterophilic datasets (Texas, Wisconsin, Actor, Squirrel, and Chameleon) held out for fine-tuning.

As shown in Figure A3 we find that incorporating biology datasets despite being seemingly unrelated to the target domain—improved performance on the OOD datasets. This suggests that knowledge learned from the biology domain positively impacts performance in seemingly unrelated domains. Furthermore, adding all available datasets, including synthetic graphs, boosted performance even more, indicating that diversity (not just domain specific data), is the key to improving generalization.

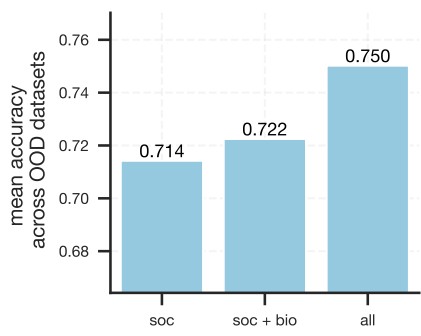

Figure A3: **Domain Scaling**: Average accuracy across OOD datasets (using MFT) for models trained on different subsets of data

### D.2 Scaling analysis breakdown for different test datasets

The main text shows the average scaling. We break down the scaling performance for different datasets (Figure A5). All of the datasets benefit from scale, with more difficult datasets benefiting more from scaling the model size and dataset.

### D.3 Ranking of different models

We visualize the ranking of the different models (Figure A6).

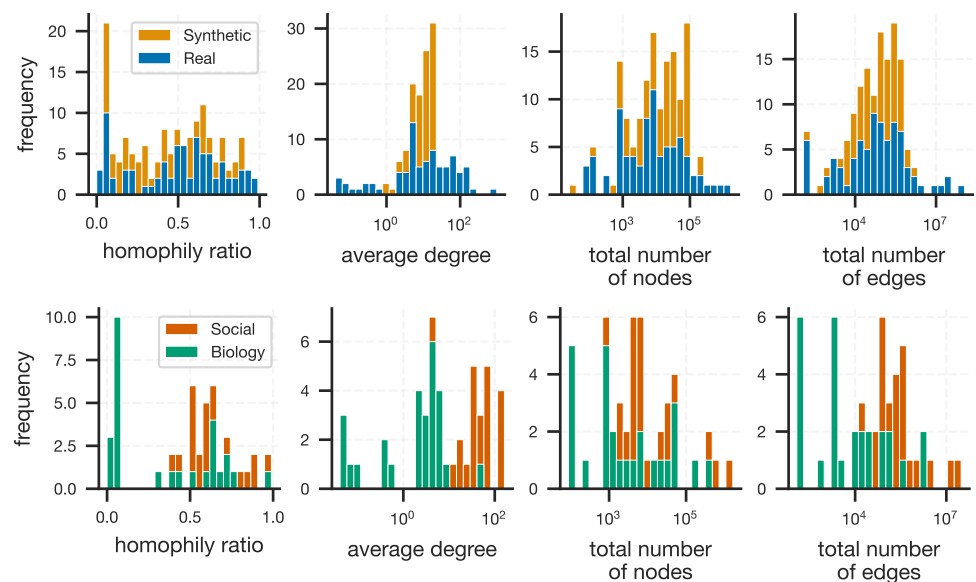

Figure A4: **Characteristics of graph datasets used to train GraphFM:** From left to right, we compute the histograms of the homophily ratio, average degree, number of nodes and number of edges of all 152 graphs used during training. The homophily ratio provides a measure of how frequently a node is directly connected to other nodes from the same class.

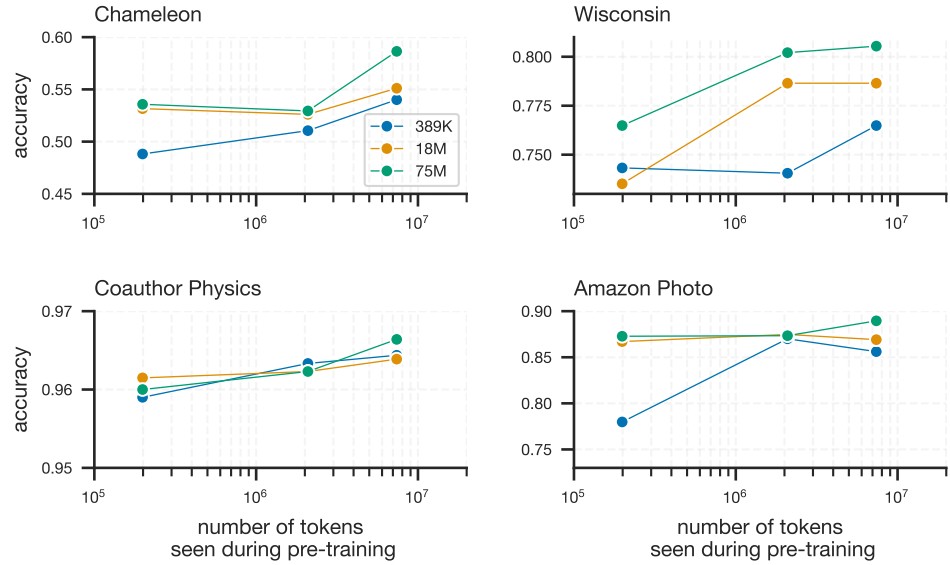

Figure A5: Accuracy as the model and dataset size are increased. Results are shown for four datasets, Chameleon and Wisconsin (heterophilic), and Coauthor Physics and Amazon Photo (homophilic).

## D.4 ADDITIONAL BASELINES

The main text presents a comparison of GRAPHFM with baselines that are more consistently reported across the literature. Table A6 and Table A7 provides additional baselines for all the OOD datasets.

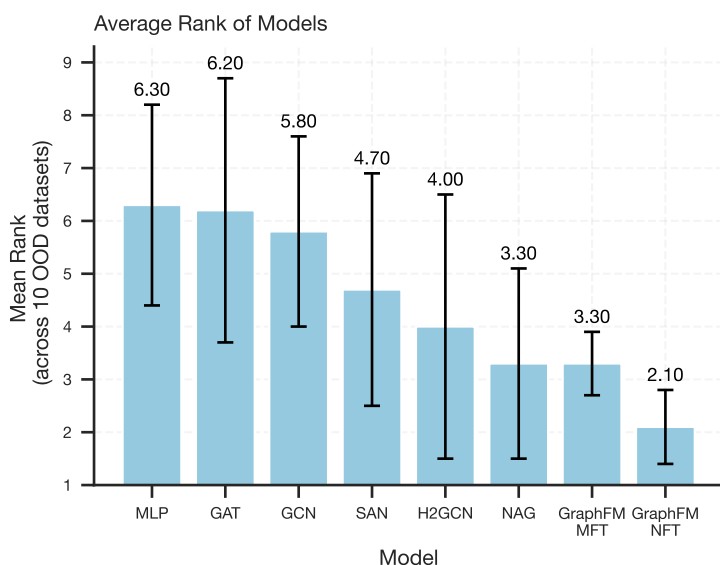

Figure A6: Mean rank of various models accross 10 OOD datasets (lower is better).

Table A6: *Results on node classification tasks for large graph datasets.* We report the accuracy (%) with standard deviation over 10 splits (OOM indicates Out of Memory).

| Method | Photo | Physics | CS | ogbn-arxiv | Comp |
|---|---|---|---|---|---|
| **GCN-based methods** | | | | | |
| GCN | 85.94±1.18 | 95.38±0.20 | 94.06±0.16 | 70.40±0.10 | 89.47 ± 0.46 |
| GatedGCN | 57.84±14.6 | 95.89±0.21 | 89.94±2.24 | 62.71±1.76 | - |
| APPNP | 84.71±1.25 | 95.04±0.31 | 87.49±0.48 | 70.20±0.16 | 90.18 ± 0.17 |
| GCNII | 67.06±1.74 | 94.88±0.32 | 84.23±0.78 | 69.78±0.16 | - |
| GAT | 87.13±1.00 | 95.14±0.28 | 93.61±0.14 | 67.56±0.12 | 90.78 ± 0.13 |
| GATv2 | 81.52±3.23 | 95.02±0.32 | 88.46±0.61 | 68.84±0.13 | - |
| SuperGAT | 85.83±1.29 | 95.11±0.26 | 88.11±0.43 | 66.99±0.07 | - |
| **Heterophily-based methods** | | | | | |
| MLP | 88.66±0.85 | 95.12±0.26 | 92.99±0.51 | 52.63±0.12 | 84.63 |
| MixHop | 93.24±0.59 | 96.34±0.22 | 93.88±0.63 | 70.83±0.30 | - |
| H2GCN | 91.56±0.70 | 96.28±0.13 | 94.02±0.31 | 68.29±0.67 | 89.33 ± 0.27 |
| FAGCN | 87.53±0.75 | 95.86±0.12 | 91.82±0.54 | 66.12±0.02 | - |
| GPRGNN | 92.27±0.44 | 96.06±0.21 | 93.60±0.36 | 68.28±0.21 | 89.32 ± 0.29 |
| **Graph Transformer-based methods** | | | | | |
| SAN | 94.17±0.65 | 96.83±0.18 | 94.16±0.36 | 69.17±0.15 | 89.83 ± 0.16 |
| Graphormer | 85.20±4.12 | OOM | OOM | OOM | OOM |
| LiteGT | - | OOM | 92.16±0.44 | OOM | - |
| UniMP | 92.49±0.47 | 96.82±0.13 | 94.20±0.34 | 73.19±0.18 | - |
| DET | 91.44±0.49 | 96.30±0.18 | 93.34±0.31 | 55.70±0.30 | - |
| NAGphormer | 94.64±0.60 | 96.66±0.16 | 95.00±0.14 | 68.21 ± 0.021 | 91.22 ± 0.14 |
| GRAPHFM -MFT | 93.01±1.82 | 96.64±0.17 | 95.19±0.21 | 65.29±0.16 | 89.95 ± 0.83 |
| GRAPHFM -NFT | 94.37±0.35 | 96.77±0.12 | 95.24±0.18 | 70.01±0.18 | 90.07 ± 0.21 |

Table A7: *Results on node classification tasks for heterophilic graphs.* We report the test accuracy across many heterophilic graph benchmark datasets. The standard deviation is reported across 10 train/test splits.

| Method | Texas | Wisconsin | Actor | Squirrel | Chameleon |
|---|---|---|---|---|---|
| **GCN-based methods** | | | | | |
| GCN | $55.14 \pm 5.16$ | $51.76 \pm 3.06$ | $27.32 \pm 1.10$ | $31.52 \pm 0.71$ | $38.44 \pm 1.92$ |
| GAT | $52.16 \pm 6.63$ | $49.41 \pm 4.09$ | $27.44 \pm 0.89$ | $36.77 \pm 1.68$ | $48.36 \pm 1.58$ |
| GraphSAGE | $82.43 \pm 6.14$ | $81.18 \pm 5.56$ | $34.23 \pm 0.99$ | $41.61 \pm 0.74$ | $58.73 \pm 1.68$ |
| **Heterophily-based methods** | | | | | |
| MLP | $80.81 \pm 4.75$ | $85.29 \pm 3.31$ | $36.63 \pm 0.70$ | $28.77 \pm 1.56$ | $46.21 \pm 2.99$ |
| HH-GCN | $71.89 \pm 3.46$ | $79.80 \pm 4.30$ | $35.12 \pm 1.06$ | $47.19 \pm 1.21$ | $60.24 \pm 1.93$ |
| HH-GAT | $80.54 \pm 4.80$ | $83.53 \pm 3.84$ | $36.70 \pm 0.92$ | $46.35 \pm 1.86$ | $61.12 \pm 1.83$ |
| HH-GraphSAGE | $85.95 \pm 6.42$ | $85.88 \pm 3.99$ | $36.82 \pm 0.77$ | $45.25 \pm 1.52$ | $62.98 \pm 3.35$ |
| MixHop | $77.84 \pm 7.73$ | $75.88 \pm 4.90$ | $32.22 \pm 2.34$ | $43.80 \pm 1.48$ | $60.50 \pm 2.53$ |
| GGCN | $84.86 \pm 4.55$ | $86.86 \pm 3.29$ | $37.54 \pm 1.56$ | $55.17 \pm 1.58$ | $71.14 \pm 1.84$ |
| H$_2$GCN | $84.86 \pm 7.23$ | $87.65 \pm 4.98$ | $35.70 \pm 1.00$ | $36.48 \pm 1.86$ | $60.11 \pm 2.15$ |
| **Graph Transformer-based methods** | | | | | |
| SAN | $60.17 \pm 6.66$ | $51.37 \pm 3.08$ | $27.12 \pm 2.59$ | $39.92 \pm 2.14$ | $44.32 \pm 1.73$ |
| UniMP | $73.51 \pm 8.44$ | $79.60 \pm 5.41$ | $35.15 \pm 0.84$ | - | - |
| ET | $56.76 \pm 4.98$ | $54.90 \pm 6.56$ | $28.94 \pm 0.64$ | - | - |
| NAGphormer | $63.51 \pm 5.85$ | $62.55 \pm 6.22$ | $34.33 \pm 0.94$ | $49.93 \pm 0.07$ | $57.39 \pm 0.02$ |
| Gapformer | $80.27 \pm 4.01$ | $83.53 \pm 3.42$ | $36.90 \pm 0.82$ | - | - |
| GRAPHFM -MFT | $80.81 \pm 2.76$ | $83.13 \pm 2.35$ | $36.29 \pm 0.63$ | $42.80 \pm 1.54$ | $58.64 \pm 1.24$ |
| GRAPHFM -NFT | $82.16 \pm 3.24$ | $83.62 \pm 3.21$ | $38.01 \pm 1.07$ | $42.98 \pm 1.62$ | $59.12 \pm 1.64$ |

