# OpenReview forum: "GraphFM: A generalist graph transformer that learns transferable representations across diverse domains"
_ICLR.cc/2025/Conference — ICLR 2025 Conference Withdrawn Submission_

### Official Review · Reviewer_o9Wd · 2024-10-25

**Soundness:** 2
**Presentation:** 3
**Contribution:** 2
**Rating:** 3
**Confidence:** 3

**Summary:**

In this paper, the authors introduce GraphFM, a transformer-based model designed for pre-training on multiple graph datasets and fine-tuning on downstream tasks. They pre-trained GraphFM on 152 diverse graph datasets and developed several strategies to improve the speed and efficiency of the pre-training process. Following pre-training, the model was fine-tuned and evaluated on various downstream graph datasets with distinct characteristics. The results demonstrate consistent performance improvements as model size and data scale increase.

**Strengths:**

1. The paper is well-written and easy to understand.

2. To my knowledge, this is the first time a single graph encoder has been trained on 152 different graph datasets and evaluated for its effectiveness—a significant and commendable achievement.

3. The evaluation of the model is comprehensive.

**Weaknesses:**

1. A primary limitation of the model is its dependency on unique initial MLPs and final predictors for each graph dataset, which necessitates fine-tuning for every new dataset or task. This requirement significantly hinders the model’s practicality in real-world applications.

2. The pre-training results were largely anticipated, given the model’s supervised training approach. However, in real-world scenarios, labeled data is often scarce, making these results difficult to scale for large-scale graph model training. Notably, what stands out about large language models (LLMs) is their ability to leverage self-supervised learning, achieving scaling laws and emergent abilities.

3. Despite extensive pre-training and fine-tuning on downstream datasets, the model’s performance only matches that of specialized models, without offering substantial improvement. Considering that the model still requires fine-tuning on new datasets, it raises questions about the benefits of this approach over simply training specialized models directly.

4. In Figure 5, the authors compare the proposed model to GCN and NAG on heterophilic graphs and the Coauthor-CS dataset. Could the authors provide comparisons between the proposed model and models with comparable performance on homophilic graphs? Additionally, how does varying the hyperparameter settings affect the performance of the proposed model?

**Questions:**

See above.

---

### Official Review · Reviewer_kJMU · 2024-10-27

**Soundness:** 3
**Presentation:** 3
**Contribution:** 2
**Rating:** 3
**Confidence:** 3

**Summary:**

GRAPHFM uses a Perceiver-based encoder to unify various graph features into a shared latent space, allowing for cross-domain adaptability. The model was trained on a large dataset covering diverse domains and evaluated on a range of classification tasks.

**Strengths:**

1. The writing is clear.

**Weaknesses:**

1. How does the model handle more challenging graph tasks, such as link prediction, given its emphasis on node classification?
2. The paper’s focus on node classification limits its scope and raises concerns about the broader applicability of the approach, especially given the minimal evaluation on other graph tasks.
3. The method is not novel. The paper is more like a technical report.

**Questions:**

See weaknesses.

---

### Official Review · Reviewer_Que9 · 2024-10-28

**Soundness:** 2
**Presentation:** 3
**Contribution:** 2
**Rating:** 5
**Confidence:** 3

**Summary:**

In this paper, the authors propose a novel scalable framework, GraphFM, for pretraining Graph Neural Networks (GNNs) across multiple graph datasets. They introduce a Perceiver-based transformer encoder to compress graph-specific details into a shared latent space, enhancing GraphFM's generalization across diverse domains. The authors propose a new sampling method called DistributedSSSampler to improve sampling efficiency in large-scale graph datasets. Experimental results demonstrate that pretraining on diverse graph structures and scaling both model size and dataset diversity helps GraphFM achieve competitive performance in node classification tasks.

**Strengths:**

- GraphFM focuses on learning transferable representations by pretraining on diverse graph datasets from various domains, which helps GraphFM to generalize well across different types of graphs without the need for tuning for each new task.
- The DistributedSSSampler proposed in GraphFM can improve the efficiency of sampling in large-scale graph learning by distributing the sampling process across multiple devices, which reduces memory bottlenecks and accelerates training.

**Weaknesses:**

- Some notations are not clearly defined. For example, the expression $\tilde{\mathbf{u}}_i=\operatorname{MLP}_g\left(\mathbf{u}_i\right)$ appears only once in the paper, and the meaning of $\tilde{\mathbf{u}}_i$ is unclear. Additionally, the calculation of the position encoding $\mathbf{p}_i$ and how $\mathbf{x}_i$ concatenates a projection of the node features are not sufficiently explained.
- The novelty of GraphFM architecture is limited. GraphFM builds upon transformer-based architectures like the Perceiver to encode graph-specific details into a shared latent space. Although this improves generalization across diverse domains, the underlying architecture does not introduce fundamentally new mechanisms for graph representation learning. A similar mechanism has also been widely used in some related research papers, such as in [1] and [2].
- The experimental results are not convincing, and the baseline methods need to be stronger. Although the authors compare the GraphFM method with some widely used baseline methods like GCN, GAT, SAN, and NAG, these are neither the latest nor the most competitive, undermining the reported performance's significance. Additionally, GraphFM does not achieve state-of-the-art (SOTA) performance in most cases.

[1] Liu C, Zhan Y, Ma X, et al. Gapformer: Graph Transformer with Graph Pooling for Node Classification[C]//IJCAI. 2023: 2196-2205.

[2] Gui A, Ye J, Xiao H. G-adapter: Towards structure-aware parameter-efficient transfer learning for graph transformer networks[C]//Proceedings of the AAAI Conference on Artificial Intelligence. 2024, 38(11): 12226-12234.

**Questions:**

- What's the difference between the issue you aim to solve in this paper and the open-set graph domain adaption [3]?
- The authors should provide a theoretical analysis of the attention mechanism and position encoding in GraphFM when deployed on large-scale graphs, similar to the approach in [4].
- More recent large-scale graph learning methods should be introduced in the experimental part for fair comparisons, such as Gapformer [1], G-Adapter [2], and ATP [5].

[3] Yin N, Wang M, Chen Z, et al. DREAM: Dual structured exploration with mixup for open-set graph domain adaption[C]//The Twelfth International Conference on Learning Representations. 2024.

[4] Li H, Wang M, Ma T, et al. What Improves the Generalization of Graph Transformers? A Theoretical Dive into the Self-attention and Positional Encoding[C]//Forty-first International Conference on Machine Learning.

[5] Li X, Ma J, Wu Z, et al. Rethinking Node-wise Propagation for Large-scale Graph Learning[C]//Proceedings of the ACM on Web Conference 2024. 2024: 560-569.

---

### Official Review · Reviewer_9xop · 2024-11-03

**Soundness:** 2
**Presentation:** 2
**Contribution:** 2
**Rating:** 3
**Confidence:** 4

**Summary:**

Using graph transfromer for multiple graph training.

**Strengths:**

1. The entire paper is well presented and the authors give the details on experiments.

2. the code is available and the reprodubility should be good.

**Weaknesses:**

1. The novelty of this work is not high. It mainly uses the graph transformer cimbining with some engineering effort, like distributed tranining.

2. The paper claim that most GNN train on individual graph, which is not true. GNNs can tranin on mutiple grpah as well.

i) GPT-GNN: Generative Pre-Training of Graph Neural Networks, KDD 2020
ii) GCC: Graph Contrastive Coding for Graph Neural Network Pre-Training, KDD 2020

3. Traning one model for multiple graphs is not new, in this work, it seems that the backbone is just chaaning from GNN to Graph transformer. Any different insight?

**Questions:**

1. Traning one model for multiple graphs is not new, in this work, it seems that the backbone is just chaaning from GNN to Graph transformer. Any different insight?

2. Th baseline is also a little old.

---

### Official Review · Reviewer_uURs · 2024-11-11

**Soundness:** 2
**Presentation:** 3
**Contribution:** 2
**Rating:** 3
**Confidence:** 3

**Summary:**

The authors propose GraphFM, a pre-training approach for learning on a variety of graph datasets. The authors posit that GraphFM mitigates the need for personalization of learning graph neural networks on a particular dataset, and thus offering a scalable backbone for a variety of graph learning tasks. The paper demonstrates that GraphFM is competitive in its performance, specifically on node classification task.

**Strengths:**

The authors develop a multi-graph pretraining approach to learn GraphFM, enabling an ability to handle diverse data across a variety of domains.

**Weaknesses:**

The contributions of the paper are fairly limited, as the authors have not established the premise of developing their foundation models vis-a-vis some of the existing works, and the evaluation is also not convincing.

First, I recommend that the authors consider the survey paper, "A Survey on Self-Supervised Graph Foundation Models: Knowledge-Based Perspective," and also the tutorial on Graph Foundation Models in WWW'24. The authors have not cited the former, and also not compared and contrasted with the methods discussed in the survey paper.

Second, the authors should also look at, "Learning MLPs on Graphs: A Unified View of Effectiveness, Robustness, and Efficiency," in ICLR'24. While the paper is not focused on pre-training, but the MLP construct holds similarities to the work developed by the authors in their paper.

Third, the authors do not provide any context on why the particular data were used for validation, and not others. As such, it is fairly unconvincing, and there are also no statistical significance offered in the table of results.

Fourth, the authors have not compared their performance to heterogeneous graph neural networks (there are a number of recent papers on this topic), and hence the performance comparisons are not appropriately contextualzied, especially on heterogeneous graphs.

Fifth, why the focus only on node classification? If it is a pre-trained foundation models, then should there not be more generalizability offered on downstream tasks?

**Questions:**

Please consider the weaknesses.

---

### Note · Authors · 2024-11-25

I have read and agree with the venue's withdrawal policy on behalf of myself and my co-authors.